# OFFLINE FEDERATED DEEP REINFORCEMENT LEARNING WITH AWARENESS OF EXPECTED RETURNS AND POLICY INCONSISTENCY

## ABSTRACT

Offline Federated Deep Reinforcement Learning (FDRL) methods aggregate multiple client-side offline Deep Reinforcement Learning (DRL) models, each trained locally, to facilitate knowledge sharing while preserving privacy. Existing offline FDRL methods assign client weights during global aggregation using either simple averaging or Q-values, but they neglect the combined consideration of Q-values and policy inconsistency, the latter of which reflects the distributional discrepancy between the learned policy and the policy from offline data. This causes clients with no significant advantages in one aspect but obvious disadvantages in the other to disproportionately affect the global model, thereby degrading its capabilities in that aspect. During local training, clients in existing methods are compelled to fully adopt the global model, which negatively impacts clients when the global model is weak. To address these limitations, we propose a novel federated learning framework that can be seamlessly integrated into current offline FDRL approaches to improve their performance. Our method considers both policy inconsistency and Q-values to determine the weights of client models, with the latter adjusted by a scaling factor to align with the magnitude of the former. The aggregated global model is then distributed to clients, who minimize the discrepancy between their models and the global one. The impact of this discrepancy is reduced if the client's model ability exceeds that of the global model, mitigating the effect of a weaker global model. Experiments on the Datasets for Deep Data-Driven Reinforcement Learning (D4RL) demonstrate that our method enhances four state-of-the-art (SOTA) offline FDRL methods in terms of return and D4RL score.

## 1 INTRODUCTION

Offline DRL enables learning a policy from static data without prolonged environmental interactions, and is widely used in areas like recommendation systems Chen et al. (2024) and autonomous driving Lee et al. (2024a). However, a single offline DRL model often faces inefficient learning due to limited data diversity, emphasizing the need for knowledge sharing among multiple models. Offline FDRL Zhou et al. (2024a)Park & Woo (2022) addresses this by enabling distributed training and knowledge sharing across devices or edge nodes, without sharing raw data, thereby improving learning efficiency and preserving data privacy.

The main challenge in offline FDRL lies in how to aggregate client models and how to use the aggregated global model to train client models. Existing offline FDRL methods can be classified into two types. The first approach uses simple averaging to aggregate local models, as seen in Yue et al. (2024b)Yue et al. (2024a)Zhou et al. (2024a)Park & Woo (2022)Woo et al. (2024)Wen et al. (2023), but this method fails to prioritize higher-performing client models. The second approach assigns client weights in the global aggregation based on Q-values, as in Rengarajan et al. (2024), where clients with higher Q-values are given more weight in the aggregation process. However, the global aggregation and local training of existing methods have the following limitations, leading to suboptimal policies. **First, existing methods fail to comprehensively consider both policy inconsistency and Q-values when calculating client importance in global aggregation**. Since offline DRL learns a policy by fitting to offline data, maximizing the expected returns of actions and ensuring the policy's fit to offline data are equally important, and both help offline DRL learn a policy

with high long-term returns Figueiredo Prudencio et al. (2024)Levine et al. (2020). Maximizing the expected returns of actions is achieved by increasing Q-values, while improving the policy's fit to offline data requires minimizing policy inconsistency, which is the gap between the current policy and the potential policy represented by the offline dataset Figueiredo Prudencio et al. (2024)Levine et al. (2020). Thus, relying solely on Q-values for weight allocation may cause local models with no significant expected return advantage but poor data fit to occupy a larger share in the global model, severely degrading its data-fitting ability. On the other hand, focusing solely on policy inconsistency to calculate client importance neglects clients with higher expected returns, failing to effectively maximize the global model's expected return. **Second, existing methods make the local models fully adopt the global model's knowledge during local training**. Since there is no guarantee that the global model's performance will always be better than all local models, a weak global model may negatively impact some stronger local models, degrading their performance.

Motivated by these observations, this work aims to propose a generic federated learning framework that can be seamlessly integrated into existing offline FDRL approaches to improve their performance. **To address the first limitation, we use both policy inconsistency and Q-values to determine the importance of each client**. For policy inconsistency, we use the current policy to predict actions and quantify the discrepancy between these predicted actions and those from the offline data using a distributional discrepancy metric, such as Jensen-Shannon (JS) Divergence. The Q-value is then scaled by a factor from the reciprocal of the average absolute Q-value of a mini-batch, aligning it with the magnitude of policy inconsistency to calculate client importance. Clients upload their local models and calculated importance to the server, which normalizes the importance via softmax to derive weights. The models are then aggregated with these weights to form the global model, which is sent back to the clients for local training. **To address the second limitation, we reduce the impact of a weaker global model on the local models**. During local training, each client minimizes the discrepancy between its local model and the global model to learn from the latter. In this process, we assess the abilities of both the global and local models using the same method employed in global aggregation, which combines policy inconsistency and Q-value. If a client's ability exceeds that of the global model, the influence of the discrepancy is reduced with a decay factor, mitigating the negative effects of a weaker global model on local model updates.

The key contributions are summarized as follows: 1) We introduce a novel, generic federated learning framework that considers both policy inconsistency and Q-values to calculate the importance of each client in global aggregation, and reduces the influence of a weak global model on stronger local models during local training; 2) We present a complexity analysis of our method, as well as a theoretical analysis to explain the superior performance of our approach compared to existing methods; 3) Extensive experiments demonstrate that our method improves four SOTA offline FDRL methods on the D4RL dataset, achieving higher returns and D4RL scores.

## 2 RELATED WORK

**Offline DRL**. Offline DRL learns a policy from a static dataset generated by a behavior policy, reducing the need for extensive environment interactions Figueiredo Prudencio et al. (2024)Levine et al. (2020). A key challenge is minimizing policy inconsistency, where the policy must reduce the gap between its state-action distribution and that of the offline dataset Figueiredo Prudencio et al. (2024)Levine et al. (2020). Solutions such as regularization Fujimoto & Gu (2021), data rebalancing Jiang et al. (2023)Yu et al. (2022)Hong et al. (2023), and weighted behavior cloning Peng et al. (2023)Liu et al. (2024) have been proposed to address this. However, conventional offline DRL is limited to training on a fixed dataset, restricting the model's ability to learn beyond it.

**Federated Learning (FL)**. Federated Learning (FL) is a distributed machine learning approach where clients collaboratively train a shared model without exchanging data, ensuring privacy. FedAvg is one example that enables this collaboration without data sharing McMahan et al. (2017)Lee et al. (2024b). A challenge in FL is data heterogeneity, which can slow down and destabilize convergence Karimireddy et al. (2020)Wang et al. (2024)Ahmed et al. (2024). To address this, methods such as FedProx Karimireddy et al. (2020) have been proposed to reduce the gap between local and global models. Recently, FL has been applied to fine-tune large models Wu et al. (2024)Liu et al. (2025)Ye et al. (2024). While FL focuses on supervised learning with local client data, offline FDRL aims to train a policy using offline samples, targeting a policy better than the one represented by those

samples. This requires the development of aggregation strategies for offline FDRL that differ from traditional FL to address data heterogeneity effectively.

**FDRL**. FDRL trains DRL models in a federated learning manner, ensuring privacy while enabling knowledge sharing. Previous research focused on online FDRL, including Horizontal Federated Deep Reinforcement Learning (HFDRL) and Vertical Federated Deep Reinforcement Learning (VFDRL). HFDRL involves independent agents in different environments Cha et al. (2020)Jiang et al. (2025)Wang et al. (2023), while VFDRL emphasizes collaboration in a shared environment with limited observations Zhuo et al. (2019). Due to inefficiencies in prolonged online interactions, offline FDRL has emerged, training multiple offline client DRL models with local static datasets. Existing offline FDRL methods fall into two groups: one uses simple averaging to aggregate client models Yue et al. (2024b)Yue et al. (2024a)Zhou et al. (2024a)Park & Woo (2022)Woo et al. (2024)Wen et al. (2023), which ignores client heterogeneity, and the other uses Q-values Rengarajan et al. (2024) to calculate client weights. However, existing offline FDRL methods have two limitations. First, they fail to comprehensively consider both policy inconsistency and Q-values when calculating client importance, which leads to clients with no significant advantage in one aspect but obvious disadvantages in the other, impairing the global model in that aspect. Second, they force clients to fully adopt the global model, which harms stronger clients when the global model is weak.

## 3 PRELIMINARIES

**FL**. FL aggregates models from various clients into a global model $\theta$ without sharing local device data. The aggregated model is then redistributed to each client for local training. Let $N_a$ be the total number of clients, with each client operating a local model. Let $\mathcal{L}_i$ represent the loss over the local data for the $i$-th client model $\theta_i$, and $w_i$ represent the weight assigned to the $i$-th client. The objective of FL is to minimize the loss function $\mathcal{L}$, which is the weighted aggregation of individual clients' losses. This can be expressed as: $\min_\theta \mathcal{L} = \min \sum_{i=1}^{N_a} w_i \mathcal{L}_i(\theta_i)$. The aggregation method, including how $w_i$ is computed, is crucial in FL. For example, $w_i$ can be determined by the ratio of $n_i$ (the sample batch size for the $i$-th client) to the total sum of all $n_i$ values. Recently, FL has introduced several advanced model aggregation techniques, such as FedProx Yuan & Li (2022), FedCAda Zhou et al. (2024b), and FedAdam Ju et al. (2024).

**Offline DRL**. DRL is a learning method based on the Markov Decision Process (MDP), defined as $\mathcal{M} = \langle \mathcal{S}, \mathcal{A}, r, \mathbb{P}, \gamma \rangle$, which includes the state space $\mathcal{S}$, action space $\mathcal{A}$, reward function $r$, transition dynamics $\mathbb{P}$, and discount factor $\gamma$. Unlike conventional DRL, which obtains samples through interaction with the environment, offline DRL aims to learn a policy $\pi$ using a static dataset $\mathcal{D}$ that contains transitions $(s, a, r, s')$ without further interaction. Each tuple $(s, a, r, s')$ in $\mathcal{D}$ represents the state, action, reward, and next state, collected by a behavior policy $\pi_b$. The goal of offline DRL is to learn a policy $\pi(s)$ that maximizes the long-term reward $J(\pi(s))$ over the static dataset: $J(\pi(s)) = \max \mathbb{E}_{\pi(s)} \left[ \sum_{i=t}^{T} \gamma^{i-t} r_i \mid s_0, a_0 \right]$ where $T$ is the learning duration, and $t$ starts at 0. In offline DRL, since the dataset $\mathcal{D}$ is pre-collected through another policy, the agent samples a mini-batch $\mathcal{D}_0$ from $\mathcal{D}$ at each step to update the model. This mini-batch has a different distribution from the current actor being updated, represented by the difference between the actor's predicted actions $\hat{a} = \pi(s; \theta^\mu)$ and the offline actions $a$. This difference, referred to as policy inconsistency, is minimized in offline DRL. In DRL, the actor-critic method is a widely used framework comprising two key components. The actor, denoted as $\theta^\mu$, parameterizes the policy $\pi(s; \theta^\mu)$, while the critic, represented by $\theta$, parameterizes the Q function $Q(s, a; \theta)$. The Q function estimates the expected cumulative reward when following the policy $\pi$, starting from state $s$ and taking action $a$. It is mathematically expressed as: $Q_\pi(s, a) = \mathbb{E}_\pi \left[ \sum_{i=t}^{T} \gamma^{i-t} r_i \mid s_t = s, a_t = a \right]$.

**Offline FDRL**. The offline FDRL implementation operates in a distributed offline setting, where $N_a$ agents collaboratively develop a policy under the guidance of a central server, without sharing raw trajectories or interacting with the environment. Each agent $i \in \{1, \ldots, N_a\}$ maintains a local dataset $\mathcal{D}_i \doteq \left\{ \left( s_{i,j}, a_{i,j}, r_{i,j}, s'_{i,j} \right) \right\}_{j=1}^{D_i}$, containing transition tuples generated by an unknown policy. The objective is to discover an optimal policy by leveraging the distributed datasets $\{\mathcal{D}_i\}_{i=1}^{N_a}$, maximizing the long-term reward. To achieve this, offline FDRL allows each client to train a local model $\theta_i$ for a certain number of epochs using its local dataset $\mathcal{D}_i$. The server then aggregates the models from

all participating clients, weighted accordingly, to obtain a global model $\theta$, which is represented as $\theta = \sum_{i=1}^{N_a} w_i \theta_i$. This global model is sent back to the clients for continued training. The process of local training followed by global aggregation is repeated until the global model is fully trained. The performance of various offline FDRL methods is assessed based on the cumulative reward from the global model, with higher values reflecting better performance.

Existing offline FDRL methods have two limitations. 1) **Inefficient global aggregation methods**. One group of methods Yue et al. (2024b)Yue et al. (2024a)Zhou et al. (2024a)Park & Woo (2022)Woo et al. (2024)Wen et al. (2023) calculates client weights $w_i$ by averaging (e.g., $w_i = \frac{1}{N_a}$), which overlooks the fact that heterogeneous clients should be assigned different weights. Another approach Rengarajan et al. (2024) assigns client weights based on Q-value size, for example $w_i \propto Q_i$, but it neglects policy inconsistency. This is likely to cause clients with no significant advantages in Q-values but poor data-fitting abilities to occupy a larger share in global aggregation and severely degrade its data-fitting ability. 2) **Inefficient local training methods**. During local training, existing methods force clients to fully adopt the global model's experience, causing a weak global model to negatively impact local models.

## 4 METHODOLOGY

This section outlines the framework and details, followed by a complexity analysis of our method.

### 4.1 THE FRAMEWORK FOR OUR METHOD

Fig. 1 illustrates the framework of our method, which includes **local model training** and **global aggregation**. During the global aggregation phase, each client samples a mini-batch from its local dataset and computes the policy inconsistency using both the actions predicted by the policy and the offline actions. This is combined with the Q-value to determine each client's importance, which, along with the local model, is uploaded to the server. The server computes the weight of each client using softmax normalization based on importance and performs a weighted sum of their models to obtain the global model. In the local training phase, the global model is sent to each client, where it minimizes the discrepancy between local and global models. A decay factor is applied to reduce the impact of the global model on local updates when the global model's performance, as evaluated through policy inconsistency and Q-value, is weaker than the local model's performance. Once local training is complete, global aggregation occurs again, repeating this process until training ends.

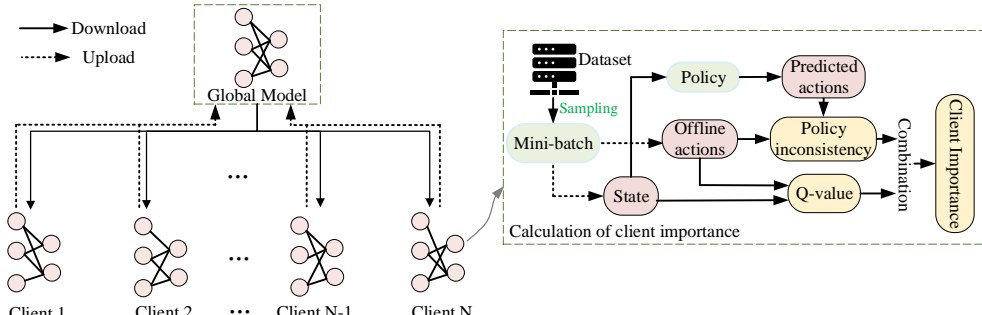

Figure 1: The framework of our approach. Here, $N$ clients participate in federated learning, with each client calculating its own importance in the same manner.

### 4.2 GLOBAL AGGREGATION

After local training on each client for a specified number of epochs, the models are uploaded to the server for aggregation. Let $Q_{\theta^{Q_i}}(s, a)$ represent the Q-value for the $i$-th client, typically generated by the first critic in the local offline DRL model, $\theta_i^\mu$ the actor for the $i$-th client, and $a_i$ the action from the $i$-th client's mini-batch $\mathcal{D}_{0i}$, which is sampled from the static dataset $\mathcal{D}_i$. The importance assessment for each client considers both the Q-value and the policy inconsistency, $Dis(\theta_i^\mu(s), a_i)$.

The importance $I_i$ for the $i$-th client is defined as: $I_i = \mathbb{E}_{(s,a)\sim\mathcal{D}_{0i}}[\kappa_i Q_{\theta^{Q_i}}(s,a) - Dis(\theta_i^\mu(s), a_i)]$. Here, $Dis(\theta_i^\mu(s), a_i)$ can be computed using any distributional difference metric, such as the squared difference: $Dis(\theta_i^\mu(s), a_i) = (\pi_{\theta_i^\mu}(s) - a_i)^2$. The server aggregates local models as follows.

**(1) Calculating Policy Inconsistency**: Besides the basic squared difference, advanced distribution discrepancy measures such as the JS divergence, a robust method for assessing distribution differences and resistant to outliers, can be used. Specifically, we first compute the difference between the predicted and offline actions, $\pi_{\theta_i^\mu}(s) - a_i$, and map this difference to a multivariate Gaussian distribution $\mathcal{N}(\mu, \Sigma)$. The multivariate Gaussian is ideal for modeling complex action distributions in DRL within high-dimensional spaces Williams (1992) Todorov et al. (2012) Nasiriany et al. (2021) Hollenstein et al. (2022). The mean and covariance $(\mu, \Sigma)$ are given by: $\mu = \frac{1}{|\mathcal{D}_{0i}|}\sum_{(s,a)\in\mathcal{D}_{0i}}(\pi_{\theta_i^\mu}(s) - a_i)$; $\Sigma = \frac{1}{|\mathcal{D}_{0i}|-1}\sum_{(s,a)\in\mathcal{D}_{0i}}\left(\pi_{\theta_i^\mu}(s) - a_i - \mu\right)^\top\left(\pi_{\theta_i^\mu}(s) - a_i - \mu\right)$. Next, we compute the JS divergence to evaluate $Dis(\theta_i^\mu(s), a_i)$ between the distribution $\mathcal{N}(\mu, \Sigma)$ and the standard multivariate Gaussian distribution $\mathcal{N}(\mathbf{0}, \sigma\mathbf{I})$, where $\mathbf{I}$ is the identity matrix and $\sigma = 0.15$ is a commonly used value, as follows: $Dis(\theta_i^\mu(s), a_i) = \text{JSD}\left(\mathcal{N}(\mu, \Sigma)|\mathcal{N}(\mathbf{0}, \sigma\mathbf{I})\right) = \frac{1}{2}\text{KLD}\left(\mathcal{N}(\mu, \Sigma)\|\frac{\mathcal{N}(\mathbf{0},\sigma\mathbf{I})+\mathcal{N}(\mu,\Sigma)}{2}\right) + \frac{1}{2}\text{KLD}\left(\mathcal{N}(\mathbf{0}, \sigma\mathbf{I})\|\frac{\mathcal{N}(\mathbf{0},\sigma\mathbf{I})+\mathcal{N}(\mu,\Sigma)}{2}\right)$ where $\text{JSD}(\cdot\|\cdot)$ denotes the JS Divergence (JSD) and $\text{KLD}(\cdot\|\cdot)$ represents the Kullback-Leibler Divergence (KLD).

**(2) Calculating Client Importance**: Here, we mainly calculate $\kappa_i$, which balances $(\pi_{\theta_i^\mu}(s) - a_i)^2$ and $Q_{\theta^{Q_i}}(s,a)$, and is given by: $\kappa_i = \frac{1}{\frac{1}{|\mathcal{D}_{0i}|}\sum_{(s_i,a_i)\sim\mathcal{D}_{0i}}|Q(s_i,a_i)|}$. The term $\kappa_i$ is necessary because each client clips actions to the range of [-1, 1], ensuring that $Dis(\theta_i^\mu(s), a_i)$ does not become too large, which could significantly differ in magnitude from $Q_{\theta^{Q_i}}(s,a)$. For instance, when using the squared difference, $(\pi_{\theta_i^\mu}(s) - a_i)^2$, $Dis(\theta_i^\mu(s), a_i)$ can be at most 4. Therefore, we use the inverse of the average absolute Q-values over a mini-batch to scale $Q_{\theta^{Q_i}}(s,a)$ to a similar magnitude as $Dis(\theta_i^\mu(s), a_i)$. Then, we obtain the importance vector for the clients, $\mathbf{I} = (I_1, I_2, ..., I_{N_a})$.

**(3) Weighted Aggregation**: We normalize the importance values of $\mathbf{I}$ using the softmax operation, as each client's importance $I_i$ may differ in magnitude. The normalized values of $\mathbf{I}$ are then used as weights, $\mathbf{w} = (w_1, w_2, ..., w_{N_a})$, as follows: $\mathbf{w} = (w_1, w_2, ..., w_{N_a}) = \text{softmax}(I_1, I_2, ..., I_{N_a}) = \left(\frac{e^{I_1}}{\sum_{i=1}^{N_a}e^{I_i}}, \frac{e^{I_2}}{\sum_{i=1}^{N_a}e^{I_i}}, ..., \frac{e^{I_{N_a}}}{\sum_{i=1}^{N_a}e^{I_i}}\right)$. The resulting weight $w_i$ is positively correlated with $I_i$. A higher $I_i$ indicates greater importance of the client, thus receiving a larger weight $w_i$ in the global aggregation. Finally, we compute the aggregated global model $\theta_{global}^\mu, \theta_{global}^Q$ at the server using a weighted sum of the clients' actors and critics: $\theta_{global}^\mu = \sum_{i=1}^{N_a}w_i\theta_i^\mu; \theta_{global}^Q = \sum_{i=1}^{N_a}w_i\theta_i^Q$.

### 4.3 LOCAL TRAINING

After global aggregation, the global critics $\theta_{global}^Q$ and global actor $\theta_{global}^\mu$ are obtained. These global models are then downloaded by the clients, which aim to reduce the discrepancy between their local models and the global models, denoted as $\mathcal{L}(global, local)$, in order to learn from the global model. Since local models may outperform the global model, each client (e.g., client $i$) evaluates both the global model's performance, denoted as $I_{glo}$, and its own performance, denoted as $I_i$, by randomly sampling a mini-batch before each local update.

If $I_i > I_{glo}$, a decay factor $\beta_i$ is applied to reduce the influence of $L(global, local)$ on the local model update (e.g., client $i$), where $\beta_i^t = \beta_i^{t-1} * \zeta$, with $\beta_i$ initially set to 1 and $\zeta$ set to 0.99. Here, $t$ represents the current iteration of the local update, during which the global model's performance is weaker than that of client $i$. As the global model increasingly underperforms in more local updates, its influence on the local model continues to diminish, as indicated by $\beta_i^{t-1} * \zeta$, which essentially corresponds to a stronger penalty for the weaker global model. Conversely, if $I_i \leq I_{glo}$, the decay factor $\beta_i$ for this local update is set to 1. Each local model's critic and actor updates are as follows:

**(1) Updating Critics**: Let $Q_{\theta_{global}^{Q_j}}(s,a)$ represent the Q-value from the $j$-th global critic $\theta_{global}^{Q_j}$, and $Q_{\theta_i^{Q_j}}(s,a)$ represent the Q-value from the $j$-th critic $\theta_i^{Q_j}$ for client $i$. The loss for the $j$-th critic $\theta_i^{Q_j}$ of the $i$-th client is then given by: $\theta_i^{Q_j} = \arg\min_{\theta^Q}\mathcal{L}(\theta_i^{Q_j}), \mathcal{L}(\theta_i^{Q_j}) = \mathcal{L}_{\text{local}}(\theta_i^{Q_j}) +$

$\frac{\beta_i}{|\mathcal{D}_{0i}|} \sum_{(s,a) \in \mathcal{D}_{0i}} \left( Q_{\theta_{global}^{Q_j}}(s, a) - Q_{\theta_i^{Q_j}}(s, a) \right)^2$ where $\mathcal{L}_{\text{local}}(\theta_i^Q)$ denotes the baseline offline FDRL method's original critic loss. For example, $\mathcal{L}_{\text{local}}(\theta_i^Q)$ is typically computed as follows: $\mathcal{L}_{\text{local}}(\theta_i^{Q_j}) = \arg\min_{\theta_i^{Q_j}} \frac{1}{|\mathcal{D}_{0i}|} \sum_{(s,a) \in \mathcal{D}_{0i}} \left( y_i - Q_{\theta_i^{Q_j}}(s, a) \right)^2$ where the target Q-value $y_i$ for the $i$-th client is computed as: $y_i \leftarrow r_i + \gamma \min_{j=1,2} Q_{\theta_i^{Q_j\prime}}(s_i', \tilde{a}_i)$ where $\tilde{a}_i \leftarrow \pi_{\theta_i^{\mu\prime}}(s_i') + \epsilon$ and $\epsilon \sim \mathcal{N}(0, 0.2)$ is noise added to the target actor's actions, clipped to the range $[-1, 1]$. Here, $\gamma$ is the discount factor.

**(2) Updating Actor**: Next, we update the clients' actors. Let $\mathcal{L}_{\text{local}}(\theta_i^\mu)$ denote the original loss for the $i$-th client's actor $\theta_i^\mu$ in the baseline offline FDRL method, which is typically computed as $-\frac{1}{|\mathcal{D}_{0i}|} \sum_{(s_t, a_t) \in \mathcal{D}_{0i}} Q_{\theta_i^{Q_1}}\left( s_t, \pi_{\theta_i^\mu}(s_t) \right) + \frac{1}{|\mathcal{D}_{0i}|} \sum_{(s_t, a_t) \in \mathcal{D}_{0i}} (\pi_{\theta_i^\mu}(s_t) - a_i)^2$. The final loss for the $i$-th client's actor, $\mathcal{L}(\theta_i^\mu)$, is given by: $\pi_{\theta_i^\mu} = \arg\min_\pi \mathcal{L}(\theta_i^\mu), \mathcal{L}(\theta_i^\mu) = \mathcal{L}_{\text{local}}(\theta_i^\mu) + \frac{\beta_i}{|\mathcal{D}_{0i}|} \sum_{(s,a) \in \mathcal{D}_{0i}} \left( \pi_{\theta_i^\mu}(s) - \pi_{\theta_{global}^\mu}(s) \right)^2$ where $\pi_{\theta_i^\mu}$ represents the policy of $\theta_i^\mu$, the local update involves $\mathcal{L}(global, local)$ being defined differently for the critic and actor. For the critic, $\mathcal{L}(global, local) = \frac{1}{|\mathcal{D}_{0i}|} \sum_{(s,a) \in \mathcal{D}_{0i}} \left( Q_{\theta_{global}^{Q_j}}(s, a) - Q_{\theta_i^{Q_j}}(s, a) \right)^2$, while for the actor, it is $\frac{1}{|\mathcal{D}_{0i}|} \sum_{(s,a) \in \mathcal{D}_{0i}} \left( \pi_{\theta_i^\mu}(s) - \pi_{\theta_{global}^\mu}(s) \right)^2$.

**Complexity Analysis**: Our method has a time complexity of $O(N_a + N_a \cdot |\mathcal{D}_{0i}|)$, with a detailed computation provided in Appendix A.2, introducing minimal computational overhead to existing offline FDRL methods. Since no new model components are added or changes made to the model upload/download process, the space complexity and communication cost remain the same as with current offline FDRL methods. **Additionally, the appendix includes pseudocode (A.3) and a theoretical analysis (A.4) for our approach.**

## 5 EXPERIMENT AND ANALYSIS

This section details the experimental settings, results, and analysis. Each method is tested with five different random seeds.

### 5.1 EXPERIMENT SETTINGS

**(1) Baselines**. Four SOTA offline FDRL methods: **Federated Diffusion Q-Learning (FDQL)** Wen et al. (2023), **Federated DRL with Dual Regularization (FDRLDR)** Yue et al. (2024b), **Federated Offline Reinforcement Learning (FORL)** Yue et al. (2024a), and **Federated Ensemble-Directed Offline Reinforcement Learning Algorithm (FEDORA)** Rengarajan et al. (2024).

**(2) Benchmarks**. We use the D4RL dataset Fu et al. (2020) as a benchmark, which includes four offline MuJoCo tasks: HalfCheetah, Hopper, Walker2d, and Ant. Our setup involves 20 clients, each with a local dataset of size $|\mathcal{D}_i| = 5000$. The global model is evaluated on the expert dataset for the four MuJoCo tasks. To simulate real-world scenarios, we: 1) **Different Datasets for Clients**: 10 clients' data are sampled randomly without replacement from the D4RL expert dataset, while the other 10 clients' data are sampled randomly without replacement from the D4RL medium dataset; 2) **Clients Unaware of Dataset Quality**: Both clients and the server are unaware of dataset quality, with no access to the environment; 3) **Random Client Participation**: In each federated learning round $t$, we randomly select $N_a = 10$ clients to participate in the federated learning process. Each client performs $T = 20$ epochs of local training per round, equating to about 380 local gradient steps.

**(3) Metrics**. We evaluate different methods using four metrics: **the final D4RL score** (presented in the main text), **the final episode return** (presented in the main text), **the episode return** (presented in the appendix), and **D4RL score** (presented in the appendix). **The episode return** is the average reward per communication round, typically displayed as a curve showing its progression. The **D4RL score** is a metric used to evaluate offline DRL performance, utilizing the normalized reward Fu et al. (2020) and displayed as a curve showing training progress. **The final D4RL score** is the average

Table 1: Comparison with SOTA methods in terms of final episode return (Mean $\pm$ Standard deviation). Bold text indicates that our method achieves better average results than the corresponding baseline. The same presentation format is used for the other tables.

| Methods | HalfCheetah | Hopper | Walker2d | Ant |
|---|---|---|---|---|
| FDQL | $5031.07 \pm 775.86$ | $1424.06 \pm 305.45$ | $3065.37 \pm 948.57$ | $3028.94 \pm 1453.11$ |
| **Ours+FDQL** | **5747.2** $\pm 1180.18$ | **1621.01** $\pm 442.34$ | **3566.6** $\pm 613.21$ | **3127.13** $\pm 1356.95$ |
| FDRLDR | $5716.45 \pm 534.46$ | $1426.13 \pm 476.2$ | $3407.13 \pm 651.35$ | $2428.22 \pm 952.39$ |
| **Ours+FDRLDR** | **5838.47** $\pm 723.02$ | $1382.66 \pm 336.58$ | **3467.93** $\pm 700.12$ | **2959.29** $\pm 1029.58$ |
| FORL | $4810.62 \pm 1083.0$ | $1506.8 \pm 349.45$ | $2895.01 \pm 618.22$ | $2163.45 \pm 1311.0$ |
| **Ours+FORL** | **6143.2** $\pm 1038.35$ | **1716.11** $\pm 642.87$ | **3292.47** $\pm 918.16$ | **2667.41** $\pm 1125.41$ |
| FEDORA | $3041.12 \pm 2279.27$ | $1588.71 \pm 381.44$ | $3122.96 \pm 840.22$ | $1537.28 \pm 931.17$ |
| **Ours+FEDORA** | **5755.13** $\pm 701.9$ | **2727.99** $\pm 1166.83$ | **5018.6** $\pm 16.67$ | **2267.79** $\pm 1212.57$ |

Table 2: Comparison with SOTA methods in terms of final D4RL score (Mean $\pm$ Standard deviation).

| Methods | HalfCheetah | Hopper | Walker2d | Ant |
|---|---|---|---|---|
| FDQL | $42.78 \pm 6.25$ | $44.38 \pm 9.39$ | $68.78 \pm 20.66$ | $79.77 \pm 34.55$ |
| **Ours+FDQL** | **48.55** $\pm 9.51$ | **50.43** $\pm 13.59$ | **77.66** $\pm 13.36$ | **82.1** $\pm 32.27$ |
| FDRLDR | $48.3 \pm 4.3$ | $44.44 \pm 14.63$ | $74.18 \pm 14.19$ | $65.48 \pm 22.65$ |
| **Ours+FDRLDR** | **49.28** $\pm 5.82$ | $43.11 \pm 10.34$ | **75.51** $\pm 15.25$ | **77.23** $\pm 25.89$ |
| FORL | $41.0 \pm 8.72$ | $46.92 \pm 10.74$ | $63.03 \pm 13.47$ | $59.19 \pm 31.17$ |
| **Ours+FORL** | **51.74** $\pm 8.36$ | **53.35** $\pm 19.75$ | **71.69** $\pm 20.0$ | **71.17** $\pm 26.76$ |
| FEDORA | $26.75 \pm 18.36$ | $49.44 \pm 11.72$ | $67.99 \pm 18.3$ | $44.3 \pm 22.14$ |
| **Ours+FEDORA** | **48.61** $\pm 5.65$ | **84.44** $\pm 35.85$ | **109.29** $\pm 0.36$ | **61.67** $\pm 28.83$ |

D4RL score from the global model over the last 10 communication rounds, while **the final episode return** is the average episode return over the same period. Higher values in these metrics are favored.

**(4) Implementation**. **The appendix A.5 presents the implementation details of our method**. The software stack required for the experiments includes Torch 1.2.0, Gym 0.16.0, and mujoco-py 1.50.0.1, whereas the hardware configuration comprises an Intel Core i7-9700 processor, 64 GB RAM, and an NVIDIA RTX 2080 GPU.

## 5.2 COMPARISON WITH SOTA OFFLINE FDRL METHODS

This section presents our improvements to four SOTA methods across four D4RL tasks: HalfCheetah, Hopper, Walker2d, and Ant, evaluated in terms of final episode return and D4RL score. The final episode return is provided in Table 1, and the final D4RL score is shown in Table 2. The results indicate that after integrating our method, by replacing the existing global aggregation methods with ours, the performance of existing offline FDRL methods improves in almost all cases, with higher episode returns and D4RL scores. This demonstrates that our global aggregation method helps existing methods achieve better performance.

Among the four baseline methods, FDRLDR, FDQL, and FORL aggregate client models by averaging with equal weight. However, due to client heterogeneity, this approach fails to prioritize higher-performing models. Similarly, FEDORA computes weights based solely on the Q-values of client models, ignoring the impact of policy inconsistency on offline FDRL performance. This may cause clients with no significant advantages in Q-values but poor data-fitting abilities to obtain higher weights in global aggregation, which severely degrades the global model's data-fitting ability and fails to fully maximize the expected returns of its policy. Meanwhile, existing methods require local models to fully adopt the global model, leading to weak global models negatively impacting local models. In contrast, our method integrates policy inconsistency and Q-values to compute client weights, thereby amplifying the impact of client models with better overall performance on the global model. Additionally, our method assesses both the global and local models' capabilities, mitigating the negative impact of a weak global model on local models.

## 5.3 ABLATION STUDY AND HYPERPARAMETER SENSITIVITY ANALYSIS

This section first presents an ablation study, demonstrating that 1) assessing client model importance solely based on policy inconsistency, referred to as Global Aggregation Using Policy Inconsistency (GAPI), and 2) not using a decay strategy for the influence of the global model on local models, referred to as Our Approach Without Decay (OWD) ($\zeta = 1$), are both suboptimal. The former

Table 3: Ablation study and hyperparameter sensitivity analysis (Mean $\pm$ Standard deviation).

| Methods | FDQL | FDRLDR | FORL | FEDORA |
|---|---|---|---|---|
| baseline | $42.78 \pm 6.25$ | $48.3 \pm 4.3$ | $41.0 \pm 8.72$ | $26.75 \pm 18.36$ |
| GAPI+baseline | $45.64 \pm 8.49$ | $48.8 \pm 6.27$ | $50.27 \pm 6.65$ | $46.38 \pm 6.32$ |
| OWD+baseline | $45.12 \pm 8.87$ | $45.05 \pm 6.14$ | $48.02 \pm 6.62$ | $48.53 \pm 6.39$ |
| Ours (0.8)+baseline | $45.32 \pm 9.13$ | $48.47 \pm 5.04$ | $46.84 \pm 4.57$ | $47.29 \pm 4.82$ |
| Ours (0.9)+baseline | $47.8 \pm 7.03$ | $47.78 \pm 7.32$ | $44.34 \pm 6.9$ | $45.11 \pm 6.57$ |
| **Ours+baseline** | $\mathbf{48.55} \pm 9.51$ | $\mathbf{49.28} \pm 5.82$ | $\mathbf{51.74} \pm 8.36$ | $\mathbf{48.61} \pm 5.65$ |

Table 4: Comparison with different distribution measures (Mean $\pm$ Standard deviation).

| Methods | FDQL | FDRLDR | FORL | FEDORA |
|---|---|---|---|---|
| SD+baseline | $45.66 \pm 5.41$ | $47.42 \pm 5.73$ | $49.2 \pm 8.29$ | $45.82 \pm 5.83$ |
| KLD+baseline | $43.92 \pm 6.51$ | $45.29 \pm 8.31$ | $49.76 \pm 5.27$ | $44.59 \pm 6.9$ |
| **JSD+baseline (Ours)** | $\mathbf{48.55} \pm 9.51$ | $\mathbf{49.28} \pm 5.82$ | $\mathbf{51.74} \pm 8.36$ | $\mathbf{48.61} \pm 5.65$ |

emphasizes the need to use both policy inconsistency and Q-value simultaneously to evaluate client model importance, while the latter highlights the necessity of applying a decay strategy. We then perform a sensitivity analysis on the hyperparameter $\zeta$ in our method. Two baselines are set up: 1) Ours (0.8), where the decay strategy is applied with $\zeta = 0.8$; and 2) Ours (0.9), where the decay strategy is applied with $\zeta = 0.9$. We use four SOTA methods as baselines, with HalfCheetah as the validation task. The experimental results, presented in Table 3, are based on final D4RL scores.

**First, we analyze the ablation study**, specifically comparing the integration of GAPI into existing methods (GAPI+baseline), the integration of OWD into existing methods (OWD+baseline), and our approach (Ours+baseline). The results show that GAPI+baseline produces a weaker D4RL score compared to our approach. This is because relying solely on policy inconsistency fails to identify clients with higher expected returns, represented by Q-values, preventing the global model from effectively maximizing the policy's expected returns, leading to suboptimal performance. Furthermore, without a decay strategy, as seen in the OWD+baseline method, the weaker global model negatively impacts the local models. A decay strategy helps mitigate this issue by improving local model performance, thereby enhancing the global model.

**Next, we analyze the hyperparameter sensitivity**. The results show that Ours (0.8) and Ours (0.9) perform worse compared to our method ($\zeta = 0.99$), confirming the optimality of our current hyperparameter settings. If the decay rate is too large, as in Ours (0.8) and Ours (0.9), performance deteriorates. This is because, during certain local updates, the global model may be weaker than the client model $i$, and a high decay level (e.g., $\zeta = 0.9$) reduces the global model's negative impact by adjusting $\beta_i$ to 0.9 (from an initial value of 1). If the global model weakens further, $\beta_i$ decreases further (e.g., to 0.81), drastically reducing the global model's influence on the local model. However, due to the inherent instability of DRL model performance Xu et al. (2025b)Xu et al. (2025a), where the global model may be weaker than the local model during some updates, this does not imply that the global model always provides negative experiences or consistently performs worse than the local model. Therefore, an excessively large decay level reduces the potential for the global model to positively influence local model updates, leading to a suboptimal global model. Based on these results, the decay level should not be too large (e.g., using a very small $\zeta$).

## 5.4 COMPARISON WITH DIFFERENT DISTRIBUTION MEASURES

This section evaluates our method's performance using various metrics to assess policy inconsistency, focusing on three indicators: **Squared Difference (SD)**, **KLD**, and **JSD**. The experimental results presented in Table 4 analyze the final D4RL score, with HalfCheetah as the validation task. Our results indicate that the method achieves optimal performance when employing JSD to measure policy inconsistency. Unlike SD, which is based on Euclidean geometry, JSD uses a multivariate Gaussian distribution to capture differences among actions, making it particularly suitable for high-dimensional actions common in DRL tasks, such as those in MuJoCo. Furthermore, JSD provides a more sophisticated approach than KLD, as it mitigates the asymmetry inherent in KLD, ensuring a more reasonable assessment of policy inconsistency. As a result, both qualitative and quantitative analyses have led us to select JSD as our preferred metric for measuring policy inconsistency.

Table 5: Comparison under different federated learning configurations (Mean $\pm$ Standard deviation).

| | Federation with varying proportions of medium participants | | | |
|---|---|---|---|---|
| Methods | FORL (25% medium) | FDRLDR (25%) | FORL (75% medium) | FDRLDR (75%) |
| Baseline | $62.94 \pm 13.33$ | $59.23 \pm 16.62$ | $41.86 \pm 5.99$ | $45.25 \pm 4.37$ |
| **Ours + baseline** | $\mathbf{63.92} \pm 15.47$ | $\mathbf{64.71} \pm 13.9$ | $\mathbf{43.59} \pm 6.93$ | $\mathbf{45.87} \pm 3.31$ |
| | Different numbers of local training epochs | | | |
| Methods | FORL (10) | FDRLDR (10) | FORL (30) | FDRLDR (30) |
| Baseline | $38.28 \pm 10.13$ | $41.86 \pm 6.5$ | $41.46 \pm 7.11$ | $43.29 \pm 7.94$ |
| **Ours + baseline** | $\mathbf{40.26} \pm 7.27$ | $\mathbf{44.68} \pm 5.31$ | $\mathbf{46.1} \pm 9.24$ | $\mathbf{48.04} \pm 6.56$ |
| | More clients with the fixed proportion of aggregation participants | | | |
| Methods | FDRLDR (15:30) | FORL (15:30) | FDRLDR (20:40) | FORL (20:40) |
| Baseline | $47.79 \pm 6.29$ | $38.72 \pm 9.49$ | $51.05 \pm 5.57$ | $46.45 \pm 7.58$ |
| **Ours + baseline** | $\mathbf{50.8} \pm 6.42$ | $\mathbf{49.41} \pm 9.4$ | $\mathbf{52.78} \pm 8.41$ | $\mathbf{52.93} \pm 4.44$ |
| | Different proportions of aggregation participants | | | |
| Methods | FDRLDR (5:20) | FORL (5:20) | FDRLDR (15:20) | FORL (15:20) |
| Baseline | $48.67 \pm 10.02$ | $45.13 \pm 10.39$ | $43.88 \pm 6.32$ | $42.98 \pm 7.58$ |
| **Ours + baseline** | $\mathbf{51.57} \pm 6.53$ | $\mathbf{48.52} \pm 4.36$ | $\mathbf{48.62} \pm 3.69$ | $\mathbf{45.69} \pm 8.75$ |
| | Different client dataset sizes | | | |
| Methods | FDRLDR (2500) | FORL (2500) | FDRLDR (10000) | FORL (10000) |
| Baseline | $28.9 \pm 8.14$ | $28.04 \pm 7.77$ | $79.81 \pm 7.39$ | $78.18 \pm 10.67$ |
| **Ours + baseline** | $\mathbf{33.44} \pm 9.92$ | $\mathbf{31.3} \pm 8.76$ | $\mathbf{83.88} \pm 7.91$ | $\mathbf{82.45} \pm 8.78$ |

## 5.5 COMPARISON UNDER DIFFERENT FEDERATED LEARNING CONFIGURATIONS

This section compares our method with existing approaches across five different federated learning configurations to further demonstrate its superiority. **First**, we explore federation with varying proportions of medium client participants, adjusting the proportion of clients using the medium dataset to 25% and 75%. **Second**, we examine different numbers of local training epochs by changing the local training duration after each global aggregation to 10 and 30 epochs. **Third**, we maintain a fixed proportion of 50% aggregation participating clients while varying both the total number of clients and those participating in global aggregation to 30:15 and 40:20. **Fourth**, we compare the performance of different methods with client local datasets of sizes $|\mathcal{D}_i|$ set to 2500 and 10000. **Lastly**, we analyze different proportions of aggregation participants, keeping the total number of clients at 20 and setting participation ratios to 5:20 and 15:20. The experiments use FORL and FDRLDR, the two most recent methods, as baselines, with HalfCheetah as the validation task.

The experimental results for the five federated learning configurations are summarized in Table 5, showing the final D4RL scores. These results demonstrate that, even with varying configurations, such as the reduced proportion of clients utilizing the expert dataset as shown in Table 5, our method consistently improves SOTA offline FDRL methods, further validating its effectiveness. Despite these variations, our method benefits existing approaches in both global aggregation and local training designs. First, incorporating policy inconsistency and Q-values enables a more comprehensive evaluation of client significance, enhancing global aggregation. Meanwhile, reducing the interference of a weak global model on local models contributes to improving local training.

## 6 CONCLUSION

This work introduces a novel federated learning framework that can be seamlessly integrated into existing offline FDRL approaches to enhance their performance. Specifically, we consider both policy inconsistency and Q-value to compute the weights for each client model. These weighted models are aggregated into a global model, which is then distributed to clients. The clients minimize the discrepancy between their models and the global model. During local updates, we reduce the impact of this discrepancy when the client's model outperforms the global model. Extensive experiments on the D4RL dataset show that our method improves four SOTA offline FDRL methods in both return and D4RL score. Future work will explore advanced metrics to measure policy inconsistency.

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

# A  APPENDIX

## A.1  THE USE OF LARGE LANGUAGE MODELS (LLMS)

This paper only uses LLMs to polish text and not for any other purpose.

## A.2  COMPLEXITY ANALYSIS

This section demonstrates the complexity introduced by the unique components of our method compared to existing offline FDRL methods. **Time complexity:** In the global aggregation phase, each client computes the Q-values and policy inconsistency for its local model, resulting in a time complexity of $O(N_a \cdot |\mathcal{D}_{0i}|)$. Then, computing each client's weight incurs a complexity of $O(N_a)$. Thus, the overall time complexity for global aggregation is $O(N_a + N_a \cdot |\mathcal{D}_{0i}|)$. In the local training phase, each client calculates the difference between the global and local models and minimizes it, leading to a time complexity of $O(N_a \cdot |\mathcal{D}_{0i}|)$. Therefore, the total time complexity is $O(N_a + N_a \cdot |\mathcal{D}_{0i}|)$, which introduces minimal additional computational cost to the existing offline FDRL methods. Lastly, since our method does not introduce new model components or alter the process of model upload and download, there is no increase in **Space complexity** or **Communication cost** compared to existing offline FDRL methods.

## A.3  THE PSEUDOCODE

The training for the server is outlined in Algorithm 1.

The training process for each client is detailed in Algorithm 2.

## A.4  THEORETICAL ANALYSIS

This section provides a theoretical analysis of how our method improves upon existing offline FDRL approaches by additionally considering policy inconsistency. We start by introducing an assumption and four lemmas to prove Theorem 1. From Theorem 1, we derive Theorem 2, which demonstrates why our method improves current approaches by incorporating policy inconsistency into global aggregation. Building on Theorem 2, we establish Theorem 3, which derives the upper bound on the performance gap between our method and the theoretically optimal global policy. Finally, using Definition 1, we derive Lemma 5, leading to Theorem 4, which highlights the difference between the Q-values of our method and the optimal global policy.

---

**Algorithm 1** Training procedure of our method (Server)

---

1: Initialize the global model $(\theta_{global}^{\mu}, \theta_{global}^{Q})$
2: Initialize the maximum number of communication rounds $E_c$ and the number of clients $N_a$.
3: Set $round = 1$
4: **repeat**
5:     **for** $k = 1$ to $N_a$ **do**
6:         Client $k$ receives the global model $(\theta_{global}^{\mu}, \theta_{global}^{Q})$
7:         Client $k$ updates its local model based on the received global model $(\theta_{global}^{\mu}, \theta_{global}^{Q})$
8:         Client $k$ computes its importance
9:     **end for**
10:    Compute client weights
11:    Aggregate the models from the clients
12:    Distribute the updated global model $(\theta_{global}^{\mu}, \theta_{global}^{Q})$
13:    $round + +$
14: **until** $E_c - round = 0$

---

**Algorithm 2** Training procedure of our method (Client)

---

1: Initialize the actor $\theta^{\mu}$, the critics $\theta^{Q_1}$ and $\theta^{Q_2}$, along with the target networks $\theta^{\mu'}$, $\theta^{Q'_1}$, and $\theta^{Q'_2}$.
2: Initialize the dataset $\mathcal{D}$, the maximum number of communication rounds $E_c$, and the maximum duration $T$ for each local training.
3: $round = 1$
4: **repeat**
5:    Decay $\beta$ if necessary
6:    **for** $t = 1$ to $T$ **do**
7:       Sample a random mini-batch $\mathcal{D}_0$ from $\mathcal{D}$ to update the model
8:       Update the critics
9:       Update the actor
10:       Every $d$ steps, update the target networks: $\theta^{Q'_j} \leftarrow \tau\theta^{Q_j} + (1 - \tau)\theta^{Q'_j}$, $\theta^{\mu'} \leftarrow \tau\theta^{\mu} + (1 - \tau)\theta^{\mu'}$, for $j = 1, 2$
11:    **end for**
12:    Each client uploads the models $\theta^{\mu}$, $\theta^{Q_1}$, and $\theta^{Q_2}$ to the server
13:    $round + +$
14: **until** $E_c - round = 0$

---

**Assumption 1**: In DRL, the reward magnitude $|r(s)|$ is limited by a constant $R_M$, such that $|r(s)| \leq R_M$.

**Definition 1**: A function is Lipschitz continuous if its rate of change remains within a fixed bound. For any two points $x$ and $y$, a constant $K$ exists such that:

$$\|f(x) - f(y)\| \leq K\|x - y\| \tag{1}$$

This implies that the function's rate of change is constrained by $K$ across the entire domain.

**Lemma 1**: The research presented in Xiong et al. (2022) indicates that the objective for DRL, $J(\pi(s))$, can be restructured as follows:

$$J(\pi(s)) = \frac{1}{1 - \gamma} \mathbb{E}_{s \sim Dis^{\pi(s)}}[r(s)] \tag{2}$$

Here, $\frac{1}{1-\gamma}$ represents a constant positive value applicable to any given policy.

**Lemma 2**: Let $Dis^{\pi(s)} : \mathcal{S} \to \mathbb{R}$ be the occupancy measure associated with policy $\pi$ Xiong et al. (2022), defined as:

$$Dis^{\pi(s)} = \int_{\mathcal{S}} \sum_{t=0}^{\infty} (1 - \gamma)\gamma^t p_0(s)p\left(s \to s', t, \pi\right) \, \mathrm{d}s \tag{3}$$

Here, $p(s \to s', t, \pi)$ denotes the probability density of transitioning from state $s$ to state $s'$ in $t$ steps, under policy $\pi$. According to the findings presented in Xiong et al. (2022), the expression for $\mathbb{E}_{s \sim Dis^{\pi(s)}}[r(s)]$ can be formulated as follows:

$$\mathbb{E}_{s \sim Dis^{\pi(s)}}[r(s)] = \int_{\mathcal{S}} r(s) Dis^{\pi(s)} \mathrm{d}s \tag{4}$$

**Lemma 3**: Let $f : S \subset \mathbb{R}^m \to \mathbb{R}$ be a function. If $p \leq q$ and the interval $[p, q]$ lies within $S$, then the inequality

$$\left| \int_p^q f(x)\mathrm{d}x \right| \leq \int_p^q |f(x)|\mathrm{d}x \tag{5}$$

holds true.

**Proof**: Let us start by considering the integrand $f(x)$. For each $x \in [p, q]$, we have:

$$|f(x)| \geq f(x) \quad \text{if} \quad f(x) \geq 0, \quad \text{and} \quad |f(x)| \geq -f(x) \quad \text{if} \quad f(x) < 0 \tag{6}$$

when we take the integral, we can directly compare the integrals of $|f(x)|$ and $f(x)$:

$$\int_p^q |f(x)| \, \mathrm{d}x \geq \int_p^q f(x) \, \mathrm{d}x \quad \text{if} \quad f(x) \geq 0,$$
$$\text{and} \quad \int_p^q |f(x)| \, \mathrm{d}x \geq -\int_p^q f(x) \, \mathrm{d}x \quad \text{if} \quad f(x) < 0 \tag{7}$$

Thus, we have:

$$\left| \int_p^q f(x) \, \mathrm{d}x \right| \leq \int_p^q |f(x)| \, \mathrm{d}x. \tag{8}$$

This completes the proof.

**Lemma 4**: The research conducted in Xiong et al. (2022) indicates that for any two actions, $\pi_1(s)$ and $\pi_2(s)$ from the set $\mathcal{A}$, there exists a positive constant $K_\pi$ such that the following relationship is satisfied:

$$\int_{\mathcal{S}} \left| Dis^{\pi_1(s)} - Dis^{\pi_2(s)} \right| \mathrm{d}s \leq K_\pi \max_{s \in \mathcal{S}} \|\pi_1(s) - \pi_2(s)\| \tag{9}$$

**Lemma 5**: In DRL, the critic, often a neural network, computes the Q-value, and current studies assume it satisfies the Lipschitz continuity condition. From Definition 1, for a constant $K_Q$, the following inequality holds:

$$\left\| Q\left(s, \pi^A(s)\right) - Q(s, \pi^B(s)) \right\| \leq K_Q \left\| \pi^A(s) - \pi^B(s) \right\| \tag{10}$$

**Theorem 1**: For two policies, $\pi^A(s)$ and $\pi^B(s)$, the difference between $J(\pi^A(s))$ and $J(\pi^B(s))$ can be bounded as follows:

$$
\begin{aligned}
&|J(\pi^A(s)) - J(\pi^B(s))| \\
&\overset{(Lemma1)}{=} \frac{1}{1-\gamma} \left| \mathbb{E}_{s \sim Dis^{\pi^A(s)}}[r(s)] - \mathbb{E}_{s \sim Dis^{\pi^B(s)}}[r(s)] \right| \\
&\overset{(Lemma2)}{=} \frac{1}{1-\gamma} \left| \int_{\mathcal{S}} r(s) \left( Dis^{\pi^A(s)} - Dis^{\pi^B(s)} \right) \mathrm{d}s \right| \\
&\overset{(Lemma3)}{\leq} \frac{1}{1-\gamma} \int_{\mathcal{S}} |r(s)| \left| Dis^{\pi^A(s)} - Dis^{\pi^B(s)} \right| \mathrm{d}s \\
&\overset{(Assumption1)}{\leq} \frac{R_M}{1-\gamma} \int_{\mathcal{S}} \left| Dis^{\pi^A(s)} - Dis^{\pi^B(s)} \right| \mathrm{d}s \overset{(Lemma4)}{\leq} \frac{R_M K_\pi}{1-\gamma} \max_{s \in \mathcal{S}} \|\pi^A(s) - \pi^B(s)\|
\end{aligned}
\tag{11}
$$

This derivation shows that the difference in long-term cumulative rewards $J$ between two policies is proportional to their discrepancy, with $\frac{R_M K_\pi}{1-\gamma}$ acting as a positive constant valid for all policies.

**Theorem 2**: Let $\pi_k(s)$ be the policy of the $k$-th local model, assuming it has the minimum policy inconsistency with its corresponding offline dataset policy $\pi_k^{off}(s)$ among all clients. The global model's policy is represented as $\pi(s) = \sum_{k=1}^{N_a} w_k \pi_k(s)$, while $\pi^*(s)$ denotes the theoretical optimal global policy. The difference between the optimal policy $J(\pi^*(s))$ and the policy learned by various offline FDRL methods $J(\pi(s))$ can be bounded as follows:

$$
\begin{aligned}
&|J(\pi^*(s)) - J(\pi(s))| \\
&= |J(\pi^*(s)) + J(\pi_k^{off}(s)) - J(\pi_k^{off}(s)) - J(\pi(s))| \\
&\leq |J(\pi^*(s)) - J(\pi_k^{off}(s))| + |J(\pi_k^{off}(s)) - J(\pi(s))| \\
&= |J(\pi^*(s)) - J(\pi_k^{off}(s))| + |J(\pi_k^{off}(s)) + J(\pi_k(s)) - J(\pi_k(s)) - J(\pi(s))| \\
&\leq |J(\pi^*(s)) - J(\pi_k^{off}(s))| + |J(\pi_k^{off}(s)) - J(\pi_k(s))| + |J(\pi_k(s)) - J(\pi(s))| \\
&\leq \frac{R_M K_\pi}{1-\gamma} \left( \max_{s \in \mathcal{S}} \|\pi^*(s) - \pi_k^{off}(s)\| + \max_{s \in \mathcal{S}} \|\pi_k^{off}(s) - \pi_k(s)\| + \max_{s \in \mathcal{S}} \|\pi_k(s) - \pi(s)\| \right)
\end{aligned}
\tag{12}
$$

**Discussion**: In this derivation, the terms $\max_{s \in \mathcal{S}} \|\pi^*(s) - \pi_k^{off}(s)\| + \max_{s \in \mathcal{S}} \|\pi_k^{off}(s) - \pi_k(s)\|$ remain constant across different offline FDRL methods. Our approach gives greater weight $w_k$ to models from clients with minimal policy inconsistency during global aggregation, resulting in a smaller difference between $\pi(s) = \sum_{k=1}^{N_a} w_k \pi_k(s)$ and $\pi_k(s)$. Consequently, $\max_{s \in \mathcal{S}} \|\pi_k(s) - \pi(s)\|$ is reduced compared to existing offline FDRL methods. This leads to a tighter upper bound on the performance difference between our method and the theoretically optimal global policy, allowing our method to more closely approach the optimal global policy. Therefore, by incorporating policy inconsistency into global aggregation, our approach improves current offline FDRL methods.

**Theorem 3**: In offline FDRL, each client's local model minimizes the policy inconsistency with the offline dataset, bounding the difference between the offline policy and the trained local policy by $\rho_1$, such that $\max_{s \in \mathcal{S}} \|\pi_k^{off}(s) - \pi_k(s)\| \leq \rho_1$. The communication frequency between global

and local models is limited, with local models undergoing global aggregation after a fixed number of epochs, which bounds the difference between each local model and the global model by $\rho_2$. Additionally, since both the offline policy and the global model's optimal policy are fixed, the difference between them is bounded by $\rho_3$. Therefore, we have $\max_{s \in \mathcal{S}} \|\pi_k(s) - \pi(s)\| \leq \rho_2$ and $\max_{s \in \mathcal{S}} \|\pi^*(s) - \pi_k^{off}(s)\| \leq \rho_3$. As a result, the upper bound for $|J(\pi^*(s)) - J(\pi(s))|$ is given by:

$$|J(\pi^*(s)) - J(\pi(s))| \leq \frac{R_M K_\pi}{1 - \gamma} (\rho_1 + \rho_2 + \rho_3) \tag{13}$$

**Theorem 4**: Let $Q\left(s, \pi^*(s)\right)$, $Q\left(s, \pi_k(s)\right)$, $Q\left(s, \pi_k^{off}(s)\right)$, and $Q(s, \pi(s))$ denote the Q-values for the global model's optimal policy $\pi^*(s)$, the $k$-th local model's policy $\pi_k(s)$, the offline policy of the $k$-th local model $\pi_k^{off}(s)$, and the global model's policy $\pi(s)$, respectively. According to Lemma 5, the upper bound on the difference between $Q\left(s, \pi^*(s)\right)$ and $Q(s, \pi(s))$ is given by:

$$
\begin{aligned}
&\|Q\left(s, \pi^*(s)\right) - Q\left(s, \pi(s)\right)\| \\
&= \left\|Q\left(s, \pi^*(s)\right) - Q(s, \pi_k^{off}(s)) + Q(s, \pi_k^{off}(s)) - Q\left(s, \pi(s)\right)\right\| \\
&\leq \left\|Q\left(s, \pi^*(s)\right) - Q(s, \pi_k^{off}(s))\right\| + \left\|Q(s, \pi_k^{off}(s)) - Q\left(s, \pi(s)\right)\right\| \\
&= \left\|Q(s, \pi^*(s)) - Q(s, \pi_k^{off}(s))\right\| + \left\|Q(s, \pi_k^{off}(s)) + Q(s, \pi_k(s)) - Q(s, \pi_k(s)) - Q(s, \pi(s))\right\| \\
&\leq \left\|Q(s, \pi^*(s)) - Q(s, \pi_k^{off}(s))\right\| + \left\|Q(s, \pi_k^{off}(s)) - Q(s, \pi_k(s))\right\| + \|Q(s, \pi_k(s)) - Q(s, \pi(s))\| \\
&\overset{(Lemma5)}{\leq} K_Q \left(\left\|\pi^*(s) - \pi_k^{off}(s)\right\| + \left\|\pi_k^{off}(s) - \pi_k(s)\right\| + \|\pi_k(s) - \pi(s)\|\right) \\
&\leq K_Q (\rho_1 + \rho_2 + \rho_3)
\end{aligned}
$$

$$\tag{14}$$

### A.5 Implementation Details

Each client uses the same DRL model, which consists of a three-layer Multi-Layer Perceptron (MLP) with 256 neurons in each hidden layer and ReLU activation for both the actor and critic. The actor's output layer uses Tanh activation. Both components share a fixed learning rate of 0.001 and are optimized using the Adam optimizer. Mini-batches contain 256 samples, with a discount factor of 0.99. The target network is updated every 2 steps using a soft update rate of 0.005. We assess policy inconsistency using the JSD method, which our experiments have shown to be the most effective approach. $\zeta$ is set to 0.99.

### A.6 Comparison with SOTA offline FDRL methods

This section presents our improvements to four SOTA methods across four MuJoCo tasks in D4RL: HalfCheetah, Hopper, Walker2d, and Ant, evaluated in terms of episode return and D4RL score.

Fig. 2 shows the experimental results in terms of episode return, while Fig. 5 presents the results based on the D4RL score. The results indicate that after integrating our method, which replaces the existing global aggregation methods, the performance of offline FDRL methods improves in almost all cases, with higher episode returns and D4RL scores. This demonstrates that our global aggregation method helps existing methods achieve better performance.

### A.7 Ablation study and hyperparameter sensitivity analysis

This section first presents an ablation study, demonstrating that 1) assessing client model importance solely based on policy inconsistency, referred to as Global Aggregation Using Policy Inconsistency (GAPI), and 2) not using a decay strategy for the influence of the global model on local models, referred to as Our Approach Without Decay (OWD) ($\zeta = 1$), are both suboptimal. The former emphasizes the need to use both policy inconsistency and Q-value simultaneously to evaluate client

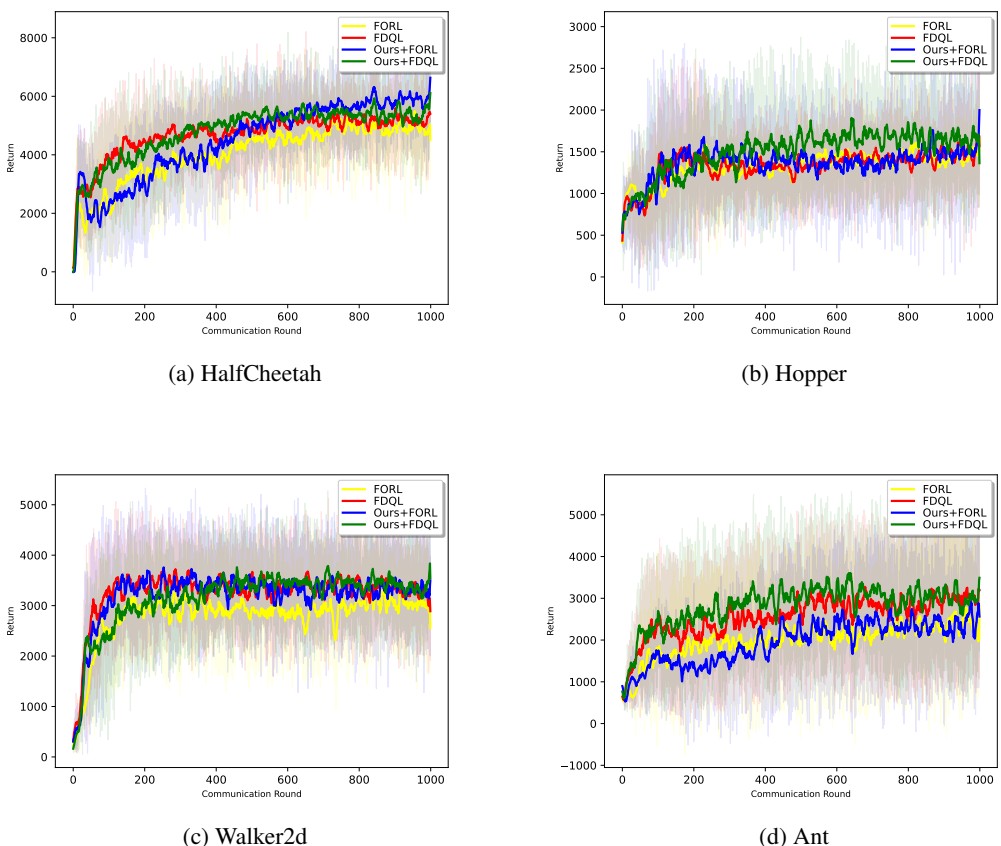

(a) HalfCheetah

(b) Hopper

(c) Walker2d

(d) Ant

Figure 2: Comparison with SOTA offline FDRL methods in terms of episode return. Here, we use FORL and FDQL as the baseline. In these figures, the x-axis represents the communication rounds, and the y-axis shows the return achieved by the server in each round. The bold curve depicts the average performance, with the shaded area indicating the standard deviation across five runs. The same format is used in the other figures.

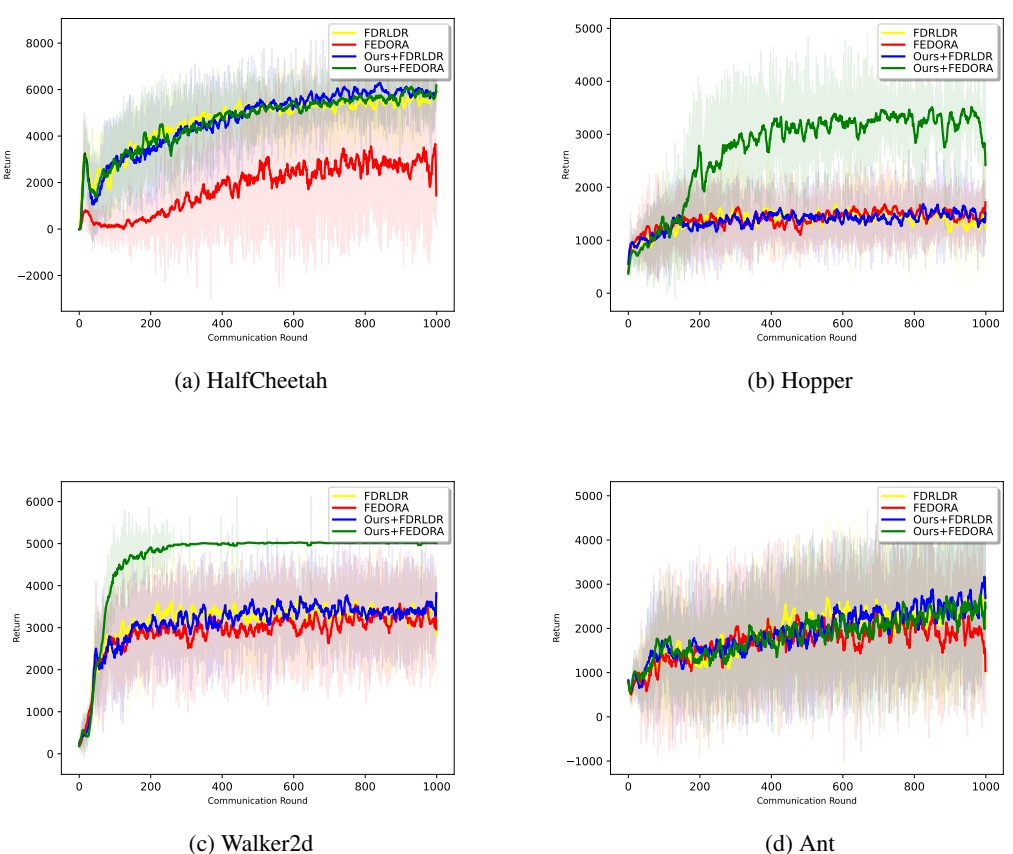

(a) HalfCheetah

(b) Hopper

(c) Walker2d

(d) Ant

Figure 3: Comparison with SOTA offline FDRL methods in terms of episode return. Here, we use FDRLDR and FEDORA as the baseline.

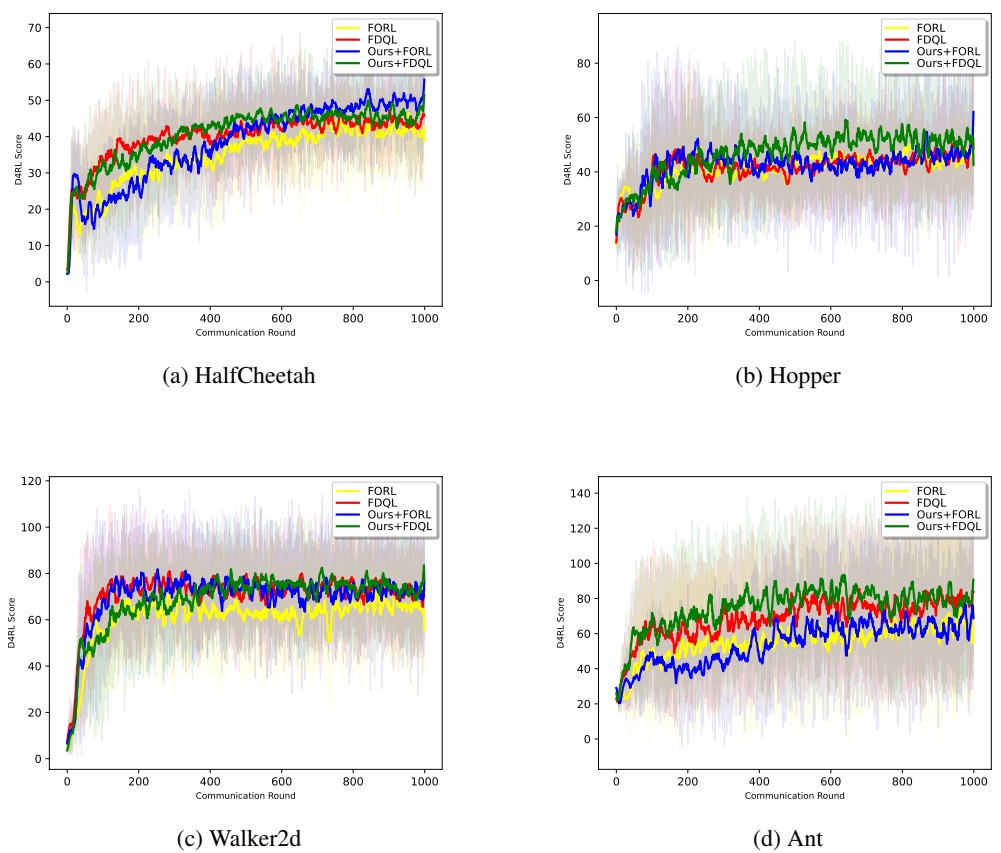

(a) HalfCheetah

(b) Hopper

(c) Walker2d

(d) Ant

Figure 4: Comparison with SOTA offline FDRL methods in terms of D4RL score. Here, we use FORL and FDQL as the baseline.

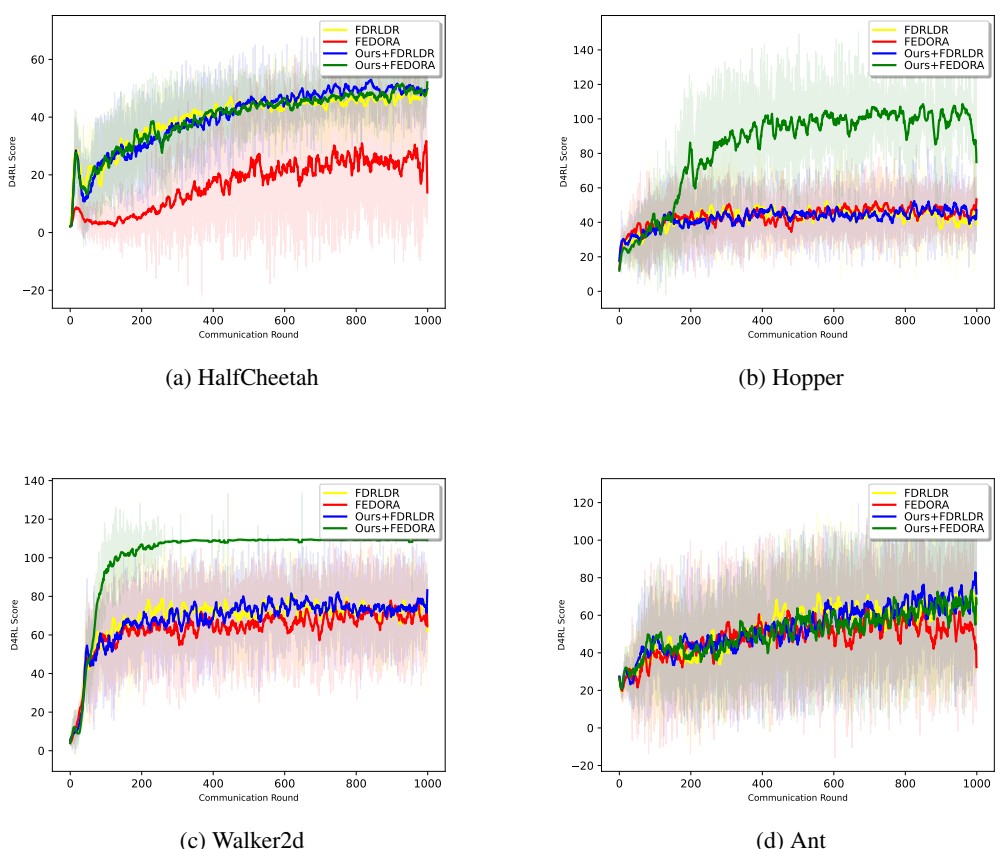

(a) HalfCheetah

(b) Hopper

(c) Walker2d

(d) Ant

Figure 5: Comparison with SOTA offline FDRL methods in terms of D4RL score. Here, we use FDRLDR and FEDORA as the baseline.

model importance, while the latter highlights the necessity of applying a decay strategy. We then perform a sensitivity analysis on the hyperparameter $\zeta$ in our method. Two baselines are set up: 1) Ours (0.8), where the decay strategy is applied with $\zeta = 0.8$; and 2) Ours (0.9), where the decay strategy is applied with $\zeta = 0.9$. We use four SOTA methods as baselines, with HalfCheetah as the validation task. The experimental results, presented in Fig. 6, are based on D4RL scores. From the experimental results, it can be seen that the full version of our method with $\zeta = 0.99$ achieves the highest D4RL scores.

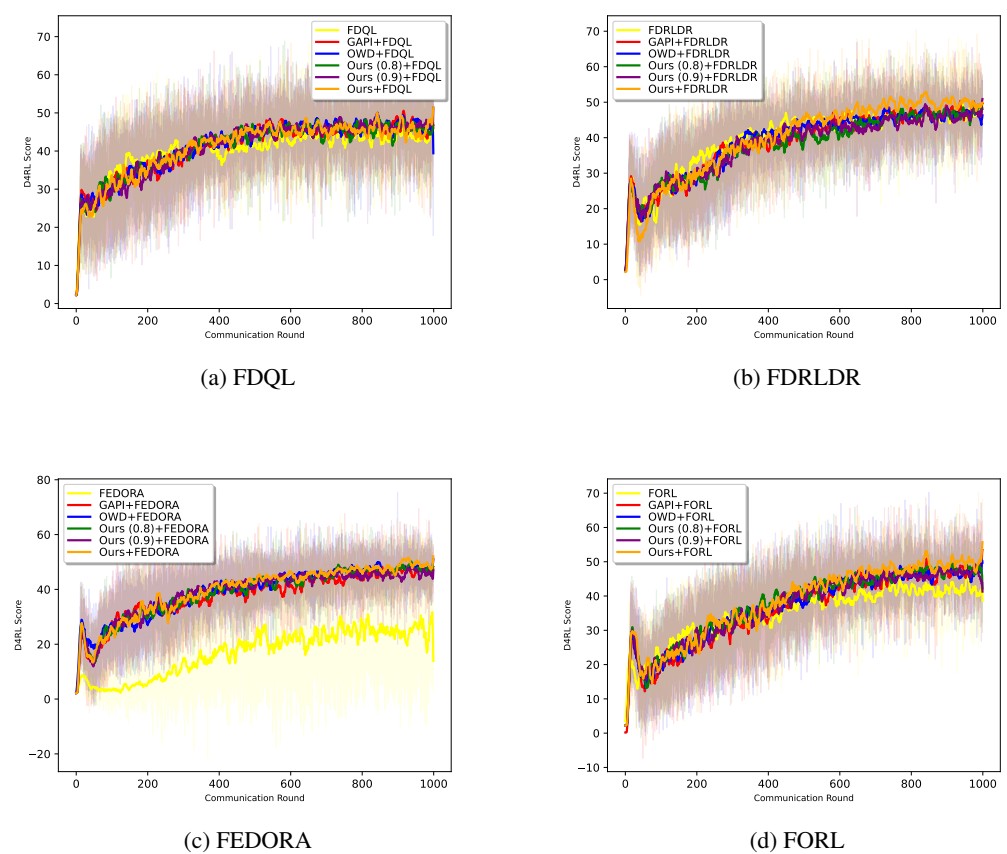

(a) FDQL        (b) FDRLDR

(c) FEDORA        (d) FORL

Figure 6: Ablation study and hyperparameter sensitivity analysis

## A.8 COMPARISON WITH DIFFERENT DISTRIBUTION MEASURES

This section evaluates our method's performance using various metrics to assess policy inconsistency, focusing on three indicators: **Squared Difference (SD)**, **KLD**, and **JSD**. The experimental results presented in Fig. 7 analyze the D4RL score, with HalfCheetah as the validation task. Our results indicate that the method achieves optimal performance when employing JSD to measure policy inconsistency.

## A.9 COMPARISON UNDER DIFFERENT FEDERATED LEARNING CONFIGURATIONS

This section compares our method with existing approaches across five different federated learning configurations to further demonstrate its superiority. **First**, we explore federation with varying proportions of medium client participants, adjusting the proportion of clients using the medium dataset to 25% and 75%. **Second**, we examine different numbers of local training epochs by changing the local training duration after each global aggregation to 10 and 30 epochs. **Third**, we maintain a fixed proportion of 50% aggregation participating clients while varying both the total number of clients and those participating in global aggregation to 30:15 and 40:20. **Fourth**, we compare the

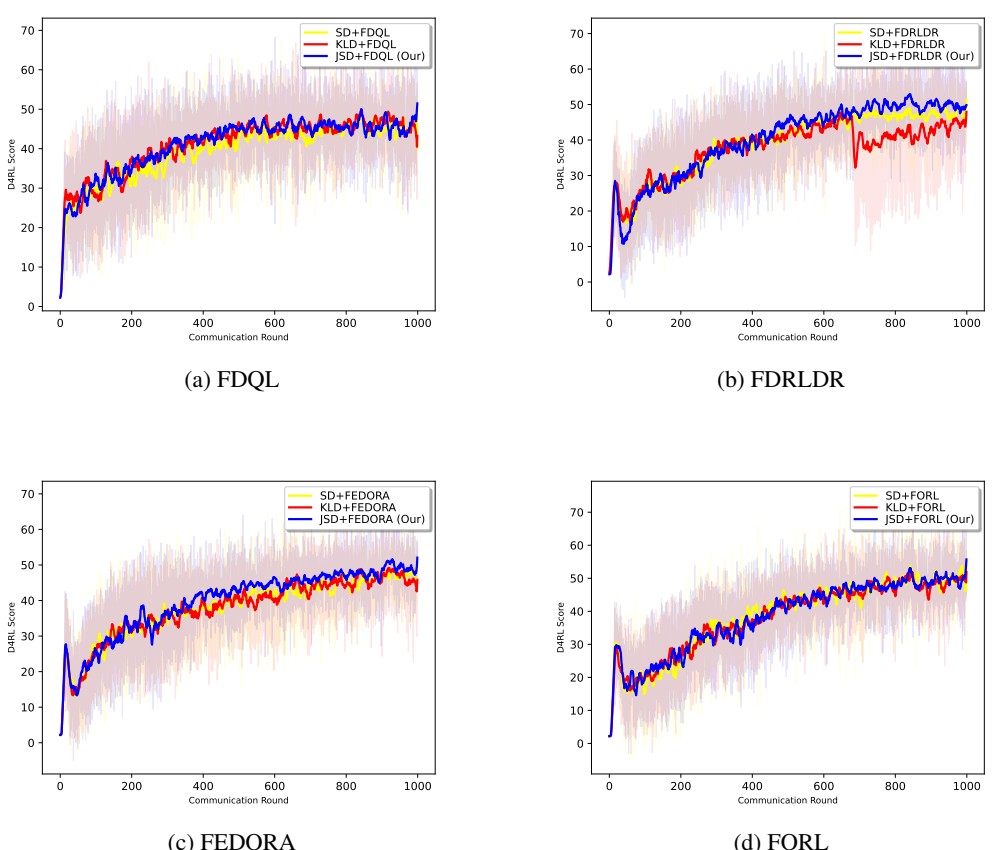

(a) FDQL

(b) FDRLDR

(c) FEDORA

(d) FORL

Figure 7: Comparison with different distribution measures

performance of different methods with client local datasets of sizes $|\mathcal{D}_i|$ set to 2500 and 10000. **Lastly**, we analyze different proportions of aggregation participants, keeping the total number of clients at 20 and setting participation ratios to 5:20 and 15:20. The experiments use FORL and FDRLDR, the two most recent methods, as baselines, with HalfCheetah as the validation task.

The experimental results for the five federated learning configurations are summarized in Fig. 8 to Fig. 12, displaying the D4RL scores. These results demonstrate that, even with varying configurations, such as the reduced proportion of clients utilizing the expert dataset as shown in Fig. 8, our method consistently improves SOTA methods, further validating its effectiveness.

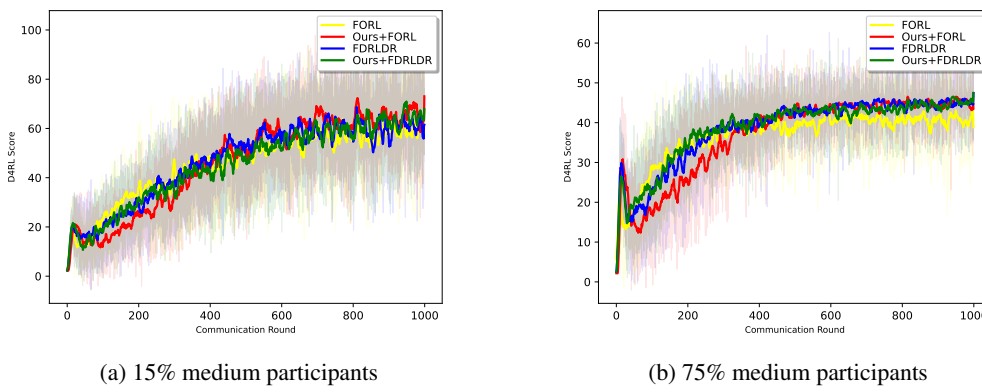

(a) 15% medium participants        (b) 75% medium participants

Figure 8: Federation with varying proportions of medium participants

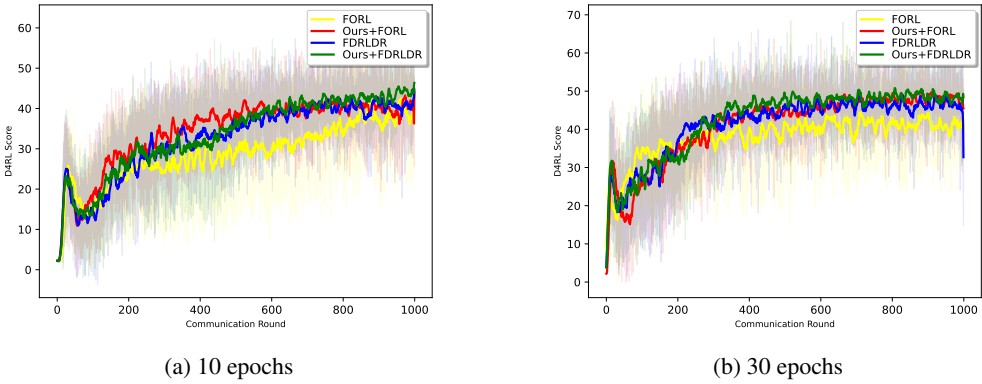

(a) 10 epochs        (b) 30 epochs

Figure 9: Different numbers of local training epochs

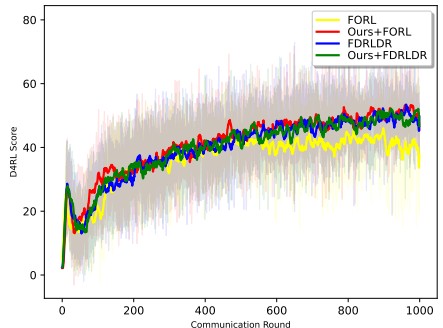 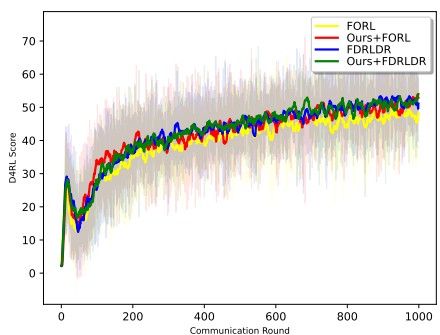

(a) A total of 30 clients, with 15 participants involved.  (b) A total of 40 clients, with 20 participants involved.

Figure 10: More clients with the fixed proportion of aggregation participants

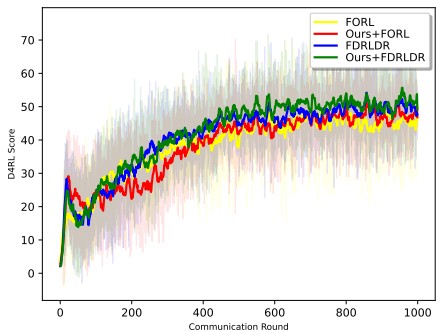 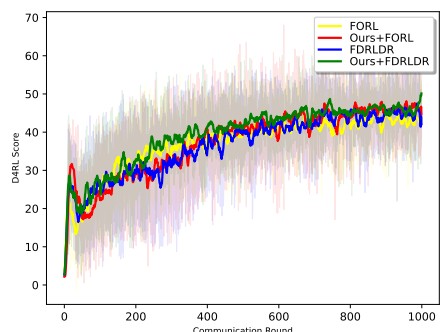

(a) A total of 20 clients, with 5 participants involved.  (b) A total of 20 clients, with 15 participants involved.

Figure 11: Different proportions of aggregation participants

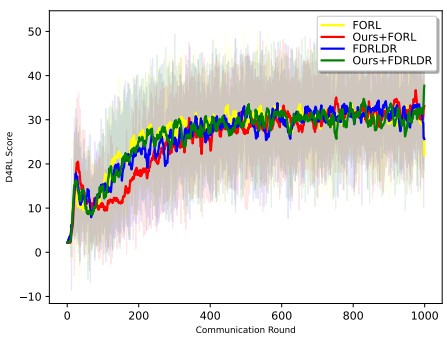 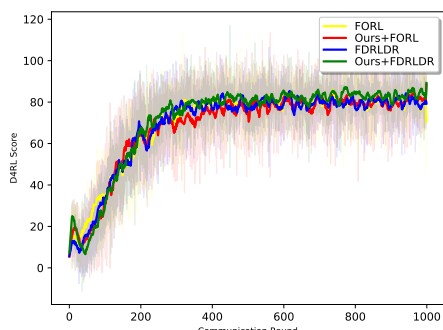

(a) The dataset size for the client is 2500.  (b) The dataset size for the client is 10000.

Figure 12: Different client dataset sizes.

