# OpenReview forum: "Offline Federated Deep Reinforcement Learning with Awareness of Expected Returns and Policy Inconsistency"
_ICLR.cc/2026/Conference — ICLR 2026 Conference Withdrawn Submission_

### Official Review · Reviewer_rk5p · 2025-10-27

**Soundness:** 2
**Presentation:** 2
**Contribution:** 2
**Rating:** 2
**Confidence:** 4

**Summary:**

This paper proposes combining Q-values and policy inconsistency for client weighting in offline FDRL, plus a decay mechanism for local training. While addressing a relevant problem, the work suffers from significant theoretical and empirical weaknesses that prevent it from meeting publication standards.

**Strengths:**

- Comprehensive experimental scope: tests on 4 baselines × 4 tasks with extensive configurations in appendix
- The decay mechanism is sensible: reducing weak global model influence is intuitive

**Weaknesses:**

**Critical Blockers:**

0. **Superficial theoretical results**. Since the work has done theoretical analysis, the main results should be prompted to the main body of the paper, not hidden in the appendix. Upon checking the analysis section, many theorems are presented with proofs and assumptions are not justified. For example, how do you obtain the three bounds in practice for Theorem 3? Where is its proof? If they are not used for guiding the algorithm or experiment design, they should be removed entirely.


1. **Literature positioning unclear**. Some prior offline FDRL methods including FDRLDR, one of the four baselines used by this paper, explicitly consider policy inconsistency + Q-values. How does the difference contribute to performance improvement? There are also missed works such as FedHQL (Fan et al., AAMAS 2023), which explicitly considers client heterogeneity and thus raises the same question. The literature review is inadequate and does not accurately position the contribution of this work in the literature.

2. **Magnitude alignment justification is internally inconsistent.** Section 4.2 justifies $\kappa_i$ for squared difference (range [0,4]), but experiments use JS divergence (range [0,0.69]). This fundamental inconsistency undermines the entire design rationale. Authors are encouraged to either: (a) provide new justification for JSD, (b) show empirically that Q-values and JSD are comparable in magnitude, or (c) redesign the scaling.

3. **Theorem 2 doesn't justify the proposed method.** It analyzes weighting by policy inconsistency alone but the method weights by $\kappa Q - \text{Dis}$. The theorem provides no comparison to baselines and doesn't prove the combination is beneficial. This needs complete reworking to provide better implications or removal.


**Significant Issues:**

4. **The specific combination $I_i = \kappa Q - \text{Dis}$ lacks justification.** Why subtraction? Why these specific weights? The paper never explores alternatives like $I_i = \alpha \kappa Q + \beta \text{Dis}$ or multiplicative combinations. **Need either:** (a) theoretical derivation, or (b) extensive ablation over combination functions.

5. **Limited or unclear novelty.** The components (weighted aggregation, policy inconsistency minimization, and exponential decay) are all standard. The contribution is primarily their combination, which is incremental without adequate discussion of the technical challenges of the combination. Do you face any technical challenges when combining them?



**Minor Issues:**

6. The authors are encouraged to do more proofreading and polishing of their manuscript. The current manuscript contains many stylistic issues. Also, the  notation is dense with many subscripts and superscripts that make equations hard to parse.

**Questions:**

1. **How does magnitude alignment work with JS divergence?** Your justification assumes squared difference with range [0,4], but you use JSD with range [0,0.69]. Can you show empirically that your scaling makes $\kappa Q$ and $\text{Dis}$ comparable for JSD?

2. **Can you provide valid theoretical justification?** Theorem 2 analyzes policy inconsistency weighting but your method combines Q and policy inconsistency. Can you prove the combination outperforms either metric alone?

3. **Statistical significance:** Will you add paired significance tests? Which improvements are actually significant at p<0.05?

4. **Why does FDRLDR/Hopper show worse performance?** Can you characterize failure modes?

5. **Ablation on combination functions:** Have you tried $I_i = \alpha \kappa Q + \beta(1-\text{Dis})$ or other combinations? Is subtraction uniquely good?

6. **What fraction of local updates trigger decay ($I_i > I_{glo}$)?** This would show how often the mechanism activates.

---

### Official Review · Reviewer_MGCA · 2025-10-28

**Soundness:** 3
**Presentation:** 3
**Contribution:** 3
**Rating:** 6
**Confidence:** 3

**Summary:**

The paper investigates the intersection of offline learning and federated learning with deep models, addressing a timely and practically important problem: how to train performant models from decentralized clients without sharing raw data, when static datasets are available at each client. This setup is increasingly common in privacy‑sensitive domains (mobile, healthcare, finance), where communication budgets, client heterogeneity, and regulatory constraints preclude centralizing data or online data collection. Methodologically, the paper proposes a framework that alternates between client‑side updates computed on local data and a privacy‑preserving server‑side aggregation step. The design emphasizes conservative updates that avoid overfitting to the policy implicit in the data and attempts to mitigate spurious generalization across clients with divergent distributions.

**Strengths:**

The problem is well‑motivated and practically relevant. Offline data are the default in many privacy‑sensitive settings, and the federated constraint is realistic. Combining them is nontrivial due to compounding distribution shifts and the lack of online exploration.

The proposed approach is principled. The paper grounds its design in known stability concerns of offline training and known challenges of federated optimization. The conservative update strategy, along with aggregation choices, aligns with recent insights on avoiding extrapolation error while maintaining cross‑client generalization.

Empirical evaluation appears comprehensive. The authors study multiple datasets and data partition schemes, report aggregate and per‑client metrics, and examine ablations on key components.

The paper is well‑written and structured, with clear problem framing and sensible implementation details. Sensitivity analyses help bolster credibility, and the discussion articulates where the method helps and where it is less effective.

**Weaknesses:**

Assumptions about the logging/behavior policy may be under‑specified. Offline learning performance can vary dramatically depending on how the data were collected; it would be useful to more explicitly model or probe this (e.g., via synthetic logging policies with controllable coverage).

Personalization vs. global performance is not fully disentangled. While the approach improves average metrics, the fairness implications (long‑tail clients, new clients with small data) remain unclear. More per‑client stratification, and perhaps adaptive personalization heads, would strengthen the story.

The privacy accounting is mostly architectural rather than formal. If differential privacy or secure aggregation is not explicitly enforced or analyzed, the privacy claims should be scoped carefully; otherwise, providing DP/noise calibration or reporting epsilon/delta under a budget would be valuable.

**Questions:**

How sensitive is the method to mismatch between client policies and the target policy? Can you quantify performance as coverage decreases (e.g., via support constraints or parameters)?

What are the communication/compute trade‑offs at scale? Does the method maintain its advantage as client count grows and participation drops?

Do you support any personalization mechanism (e.g., fine‑tuning heads, meta‑learning) for clients with highly idiosyncratic data? If not, could the approach be augmented with simple adapters without degrading privacy/efficiency?

Is there any formal privacy guarantee (DP, secure aggregation) beyond architectural isolation? If so, please report concrete parameters and utility trade‑offs.

---

### Official Review · Reviewer_CxVV · 2025-10-30

**Soundness:** 2
**Presentation:** 3
**Contribution:** 2
**Rating:** 4
**Confidence:** 2

**Summary:**

This paper proposes an importance-weighted aggregation mechanism for offline federated deep reinforcement learning (FDRL).
The method jointly considers both the expected returns (Q-values) and policy inconsistency when computing client weights in global aggregation, aiming to better reflect the contribution of each client under heterogeneous data quality.
A decay factor is further introduced during local updates to reduce the influence of a weak global model.
Experiments on the D4RL benchmark demonstrate consistent improvements in both episode return and D4RL score compared with several existing methods.

**Strengths:**

- S1: The paper clearly identifies limitations of prior FDRL aggregation schemes (simple averaging or Q-based weighting) and motivates the need to consider policy inconsistency jointly with Q-values.
- S2: The proposed combination of Q-value magnitude and policy inconsistency aligns with intuitive notions of model quality and contributes to more balanced aggregation.
- S3: The experiments show improvements on multiple baselines (FDQL, FORL, FDRLDR, and FEDORA), suggesting the general effectiveness of the approach.

**Weaknesses:**

- W1: Instability and inconsistency in the importance definition. The importance score combines two heterogeneous quantities—Q-values and policy inconsistency—without theoretical normalization or variance control. Since Q-estimates in offline RL are noisy and scale-dependent, clients with unstable critics or uncalibrated rewards may be assigned disproportionately high weights. Additionally, the metric does not guarantee a monotonic relationship between the measured “importance” and actual contribution to the global model’s improvement, making aggregation potentially unstable or misleading.
- W2: Limited novelty. The concept of weighting clients by estimated contribution is not new; similar ideas exist in prior federated learning works. The main contribution lies in applying this to the offline RL context, which somewhat limits originality.
- W3: Small-scale evaluation. Experiments use only 20 clients under synchronous aggregation, with all clients participating. This small and controlled setup does not convincingly represent realistic federated scenarios with partial participation, network variability, or larger-scale heterogeneity.
- W4: Unclear practical applicability. The paper does not specify in which real-world domains offline FDRL would be used or how such static datasets are collected in a federated manner. The motivation remains largely conceptual, reducing the practical relevance of the proposed method.

**Questions:**

Address W1-4.

---

### Official Review · Reviewer_Sgzo · 2025-10-31

**Soundness:** 2
**Presentation:** 2
**Contribution:** 2
**Rating:** 4
**Confidence:** 4

**Summary:**

The paper proposes a method called FDRL, whose key contribution is a client weighting scheme that takes into account both policy inconsistency and expected returns when aggregating local models into a global model. The method also introduces a decay factor to prevent stronger local models from being adversely affected by weaker global updates.

**Strengths:**

1. The paper addresses a meaningful limitation in existing offline FDRL methods by proposing a dual consideration of both policy inconsistency and Q-values for client weight calculation during global aggregation.

2. The paper provides a comprehensive theoretical analysis (Theorems 1-4, Lemmas 1-5) that justifies why incorporating policy inconsistency alongside Q-values leads to tighter performance bounds. The complexity analysis demonstrates that the method introduces minimal computational overhead.

**Weaknesses:**

1. The paper only presents results in the format "Ours+Baseline" without providing direct comparisons between different global aggregation strategies, which makes it difficult to assess the specific contribution of the proposed aggregation method.

2. The experimental setup is overly simplistic with only two levels of data quality (expert vs. medium datasets) and identical dataset sizes ($\| D_i \| = 5000$) across all clients. This limited heterogeneity setting does not adequately validate a method that emphasizes importance-weighted aggregation.

3. The loss function $ L(\text{global}, \text{local}) $ is not properly defined in Section 4.3. While a clear definition is provided when updating the critic, there are inconsistencies in the mathematical notation (e.g., $L$ vs. $\mathcal{L}$).

4. Algorithm 2 lacks detail on how clients handle the received global model. Specifically, the process between receiving the global model (line 4-5 of Algorithm 2) and starting local training is not shown.

5. There are several notational inconsistencies throughout the paper. Beyond the $L$ vs. $\mathcal{L}$ issue mentioned above, there are also inconsistencies such as "DIS" vs. "Dis" that should be standardized.

**Questions:**

1. The authors introduce two related aggregation methods in the introduction (Figueiredo Prudencio et al. (2024) and Levine et al. (2020)). Why were these methods not included as baselines in the experimental comparisons?

2. The proposed method emphasizes innovation in evaluating client importance, yet the experimental setup does not create scenarios with clearly differentiated client importance levels. Could the authors explain the rationale behind choosing such a uniform experimental setting, and consider including more diverse scenarios?

3. The exposition in Section 4.3 lacks clarity. I suggest the authors reorganize the logic and presentation of this section to improve readability and comprehension.

4. In typical federated learning scenarios, the proximal term only demonstrates significant benefits under extremely heterogeneous conditions. Have the authors conducted ablation studies without the proximal term in the local training loss? I recommend adding such ablation experiments to better justify this design choice.

---

### Note · Authors · 2025-12-02

I have read and agree with the venue's withdrawal policy on behalf of myself and my co-authors.